# Association of body size distortion with low body mass index in female patients with nontuberculous mycobacterial lung disease

**Yumi Takayama**[1,2]*, **Yukina Yumen**[2], **Takamasa Kitajima**[3], **Noritsugu Honda**[4], **Naoki Sakane**[5], **Motonari Fukui**[3], **Narumi Nagai**[2]

1 Department of Nutrition, Tazuke Kofukai Medical Research Institute, Kitano Hospital, Osaka, Japan, 2 Laboratory of Nutritional Physiology, Graduate School of Human Science and Environment, University of Hyogo, Hyogo, Japan, 3 Respiratory Disease Center, Tazuke Kofukai Medical Research Institute, Kitano Hospital, Osaka, Japan, 4 Department of Rehabilitation, Tazuke Kofukai Medical Research Institute, Kitano Hospital, Osaka, Japan, 5 Division of Preventive Medicine, Clinical Research Institute for Endocrine and Metabolic Disease, National Hospital Organization, Kyoto Medical Center, Kyoto, Japan

* y-takayama@kitano-hp.or.jp

## Abstract

### Background

We have previously reported that female patients with nontuberculous mycobacterial lung disease (NTM-LD) are thinner and eat less than healthy controls. Therefore, we hypothesized that their thinness is associated with body size misperception. The aim of this study was to clarify whether patients' body size perception (BSP) is associated with body mass index (BMI) independent of potential confounders.

### Methods

In this cross-sectional study, we assessed 81 outpatients for BSP using the Japanese version of Body Image Scale, body composition, dietary intake, and biochemical markers. As a control, we used BSP and self-reported anthropometric data from 111 healthy women collected via a web survey. First, BSP and BMI were compared between the patients and the controls. Second, patient data was compared between patients who overestimated their body size (OE, $n = 31$) and a control who did not (Non-OE, $n = 50$). Multiple regression analysis was performed to clarify the association between BSP (independent variable) and BMI (dependent variable), adjusting for potential confounders, such as age, disease duration, and nutritional factors.

### Results

There was a negative correlation between body size distortion and BMI in both patients and controls ($p < 0.001$, both). In interpatient comparisons, the OE group had significantly lower BMI and body fat percentage ($p < 0.001$, both), normalized energy ($p = 0.037$), and protein ($p = 0.013$) intakes, and significantly greater weight loss from age 20 ($p = 0.003$) than the Non-OE group. Multiple regression analysis revealed that overestimation of body size was

**Data Availability Statement:** Raw data cannot be shared publicly as it is in a re-identifiable database. These restrictions were placed by the Research Ethics Board of Kitano Hospital. Please contact the

Research Ethics Board of Kitano Hospital, (rinshou@kitano-hp.or.jp) for more information.

**Funding:** This study was supported by grants from the Kitano Research Incentives and Respiratory Disease Center, Tazuke Kofukai Medical Research Institute, Kitano Hospital. This work was supported in part by MHLW Comprehensive Research Project against Cardiovascular Diseases, Diabetes and Other Lifestyle Related Diseases Program Grant Number JPMH 22FA1023. The funders had no role in study design, data collection and analysis, decision to publish, or preparation of the manuscript.

**Competing interests:** The authors have declared that no competing interests exist.

associated with lower BMI independent of confounders, such as longer disease history, longitudinal weight loss, and nutritional factors.

## Conclusion

These findings suggest that body size distortion is an etiological factor for lower BMI in female patients with NTM-LD. Thus, it may be important to understand the patient's body image when providing dietary advice.

## Introduction

The prevalence of nontuberculous mycobacterial lung disease (NTM-LD), a lung infection caused by environmental bacteria, is increasing in many developed countries and regions [1–3]. Studies have reported that the incidence of NTM-LD is higher in women [2, 4, 5] and is associated with a lower body mass index (BMI) [6–8]. A previous study reported that a weight loss of $\geq 1$ kg/m$^2$ in BMI was associated with an increased incidence of NTM-LD [8]. In addition, thinness is a risk factor for poor prognosis in female patients [9–13].

We have previously shown that female patients with NTM-LD are thinner and have lower energy and protein intakes than healthy controls of the same age group, and that low body weight is associated with disease severity [14]. However, in clinical practice, we often see NTM-LD female patients who feel uncomfortable increasing their dietary intake and body weight. In order to improve the efficacy of treatment through adequate energy and nutrient intake, it is necessary to understand how NTM-LD patients perceive their body size and clarify whether their perception is disturbed.

Adult Japanese women, the background population of our study, have one of the highest prevalence rates of being underweight (11.5%) among Organization for Economic Cooperation and Development (OECD) countries [15, 16]. Intentional restriction of food intake for weight control purposes despite the ready availability of food is common among both young and middle-aged women [16]. It has been suggested that such eating behaviors are partly due to body size distortion, a misperception where the patient overestimates their own body size [17, 18]. Accordingly, we hypothesized that some female patients with NTM-LD have body size distortion similar to that of the general adult female population, and that their overestimation of body size would influence their thinness (low BMI).

To test this hypothesis, we 1) compared female patients with NTM-LD to healthy controls in terms of body size perception (BSP) and distortion, as assessed by the Japanese version of the Body Image Scale (J-BIS) developed in a previous study [19], and 2) examined whether patients' BSP (independent variable) was related to BMI (dependent variable), independent of disease duration and nutritional factors.

## Methods

### Study design and population

The study design was a single-center, cross-sectional study that included female patients with NTM-LD (age $\geq 20$ years) who visited the outpatient clinic of the Respiratory Disease Center of Kitano Hospital, Osaka, Japan, between July and October 2018. The patients enrolled in the present study were described in our previous study [14].

To compare BSP and BMI data between female patients with NTM-LD and healthy women, we used web survey data for the period from December 2020 to February 2021 as a control. These data were obtained from individuals who responded to a web-based questionnaire form (age, sex, self-reported height and weight, and body image). We selected control participants ($n$ = 111) with a similar demographic background as the patients with NTM-LD (Japanese females aged 50–75 years). Patients that had any of the following medical conditions affecting weight-related diseases within the past 5 years were excluded: cancer, eating disorders, diabetes mellitus, or endocrine disorders.

This study (participants: female patients with NTM) was approved by the Research Ethics Committee of Kitano Hospital (number: P180600300; May 1, 2018). The web-based survey study (participants: healthy women) was approved by the University of Hyogo Ethics Committee (number: 192; December 7, 2018). The study procedures were conducted in accordance with the Declaration of Helsinki and national government clinical practice guidelines. Written informed consent was obtained from all participants before enrollment.

## Evaluation of BSP

BSP was assessed using the J-BIS, which was developed and validated for the Japanese adult population [19]. The 10 silhouettes had BMI ranges (mean ± standard error) calculated from data obtained from 444 Japanese women. The mean BMIs ranged from 16.2 (Silhouette 1) to 30.2 kg/m$^2$ (Silhouette 10). Participants were asked to identify the silhouette that best represented their own body on the day of the interview or on the web survey (perceived silhouette). This BMI range was used to convert BMI to silhouette number [actual silhouette]. For those who deviated from the BMI range of each silhouette, the silhouette number of the closest BMI value was selected ($n$ = 117) or those with a BMI lower than the BMI range of silhouette 1 ($< 16.2$ kg/m$^2$) were assigned as silhouette 1 ($n$ = 10, all NTM-LD patients). Body size overestimation has been reported to be associated with weight loss [17, 18]. Overestimation was defined as a difference between perceived and actual silhouette numbers of $\geq 1$, as this difference corresponds to a $\geq 1$ kg/m$^2$ difference in BMI, a level that has been associated with the incidence of NTM-LD [8]. Accordingly, we categorized the patients into three categories: underestimation of body size (perceived silhouette–actual silhouette $< –1$), no distortion (= 0), and overestimation of body size ($< 1$). The patients were then divided into overestimation of body size (OE) and control (Non-OE) groups.

## Measurements

**Clinical measurements of patients.** Patients' medical records were reviewed for demographic and clinical data, including smoking history. On the day of the hospital visit, each patient underwent clinical measurements. Blood was sampled from the cubital vein, and the serum albumin, transthyretin (prealbumin), transferrin, retinol-binding protein, total cholesterol, hemoglobin, and lymphocyte count were measured.

**Anthropometric and nutritional assessments.** Height, weight, and body composition (InBody S10, BioSpace Co., Seoul, Korea) were measured on the same day as the interview and clinical measurements [14]. During the interview by registered dietitians, 1) maximum weight and weight at age 20 years, 2) dietary intake assessed by the 24-hour recall method for a typical weekday, 3) appetite using the Japanese version of the Simplified Nutritional Appetite Questionnaire [20], and 4) amount of physical activity as an indicator of energy expenditure using a shortened version of the International Standardized Physical Activity Questionnaire were assessed [21]. Nutritional values were calculated using computerized procedures (Excell Eiyokun, ver. 8.2, Kenpakusya Co., Japan) based on a Japanese food consumption table [22]. For

controls, self-reported height and weight data were collected, and no nutritional data were obtained.

## Sample size

Sample size was calculated using BMI as the main outcome. This calculation showed that a sample size of 26 patients per group (52 patients in total) was required to detect an inter-group difference of two points on the value of BMI (power = 0.8, alpha = 0.05) between OE and Non-OE groups (G*Power, version 3.1.9.6 for Mac [23]). This predicted difference equated to an effect size of $\geq 0.8$.

## Statistical analyses

All statistical analyses were performed using the Statistical Package for Social Sciences (SPSS for Windows™ ver. 24, IBM Inc., Tokyo, Japan). Prior to statistical evaluation, normality was tested using the Kolmogorov-Smirnov test. Comparisons between groups (NTM-LD vs. Controls or OE vs. Non-OE) were performed using unpaired Student's *t*-test or Welch's *t*-test, as appropriate. Body weight and BMI were compared by age-adjusted analysis of covariance (NTM-LD vs. Controls). Associations between BMI and body size distortion were evaluated using Spearman correlation coefficients. The receiver operating characteristic (ROC) curves were drawn to assess the prediction performance regarding the BMI and body size overestimation in the NTM-LD and control groups, and their area under the curves (AUC) were calculated using ROC curves. To compare the optimal cut-off values of BMI to best predict the body size overestimation between the two groups, the optimal cut-off values were calculated by Youden index, and the sensitivity and specificity under the optimal cut-off points in the ROC curves were obtained. The ROC analyses were performed using the R software package (version 4.0.0 for Windows, R Core Team, Vienna, Austria). Multiple regression analysis was performed to evaluate the impact of body size distortion (independent variables) on BMI (dependent variable) with adjustment for potential confounding factors (covariates), such as age, duration of NTM-LD, percentage of weight loss from 20 years of age, normalized energy intake, appetite score, and physical activity. We conducted a sensitivity analysis after excluding participants with a BMI of less than 16.2 kg/m² (all NTM-LD patients). Patients with missing data were excluded from the relevant analysis. *P* values < 0.05 were considered statistically significant.

## Results

### Anthropometric profiles and BSP in NTM-LD and controls

Table 1 shows the anthropometric profiles and BSP in NTM-LD and healthy women (controls). Compared to controls, NTM-LD were older and had a distinct anthropometric profile, including significantly lower body weight and BMI ($p < 0.001$, respectively). This difference was not significant; however, NTM-LD had overestimated their body size similar to that of the controls (38.3 vs. 27.9%, $p = 0.273$). Likewise, sensitivity analysis without < 16.2 kg/m² of BMI resulted in similar findings (S1 Table).

### Correlations between body size distortion and BMI

Fig 1 shows the correlation between body size distortion and current BMI. BMI was negatively correlated with body size distortion, whereas those with a lower BMI had a higher degree of overestimating BSP in both groups (NTM-LD, $r = -0.457$, $p < 0.001$; controls, $r = -0.475$, $p < 0.001$). Similar results ($r = -0.392$, $p = 0.001$, *Spearman*'s correlation) were obtained in a

**Table 1. Anthropometric profiles and BSP in NTM-LD and controls.**

| Variables | NTM-LD (*n* = 81) | Controls (*n* = 111) | *p* value |
|---|---|---|---|
| Age (years) [a] | 70.8 ± 8.6 | 57.1 ± 5.2 | <0.001 |
| Height (cm) [a] | 153.7 ± 6.9 | 157.9 ± 5.3 | <0.001 |
| Body weight (kg) [b] | 46.8 ± 7.2 | 55.8 ± 8.5 | <0.001 |
| Body mass index (kg/m$^2$) [b] | 19.8 ± 2.5 | 22.4 ± 3.2 | <0.001 |
| BSP [c] | | | |
| Underestimation of body size | 30 (37.0) | 44 (39.6) | |
| No distortion | 20 (24.7) | 36 (32.4) | 0.273 |
| Overestimation of body size | 31 (38.3) | 31 (27.9) | |

Mean ± standard deviation or *n* (%)

[a] Unpaired t-test

[b] analysis of covariance adjusted for age

[c] $\chi^2$ test

BSP, body size perception; NTM-LD, nontuberculous mycobacterial lung disease

Patients were divided into underestimation of body size, no distortion, and overestimation of body size based on the difference between their perceived and actual silhouettes.

sensitivity analysis excluding NTM patients with a BMI of less than 16.2 kg/m$^2$ (*n* = 71). The optimal BMI cut-off values in the NTM-LD patients (19.2 kg/m$^2$, AUC = 0.784, Sensitivity = 0.677, Specificity = 0.800) best predicting the overestimation of body size were lower than in the controls (22.3 kg/m$^2$, AUC = 0.708, Sensitivity = 0.857, Specificity = 0.500), although the sensitivity was lower in the patients.

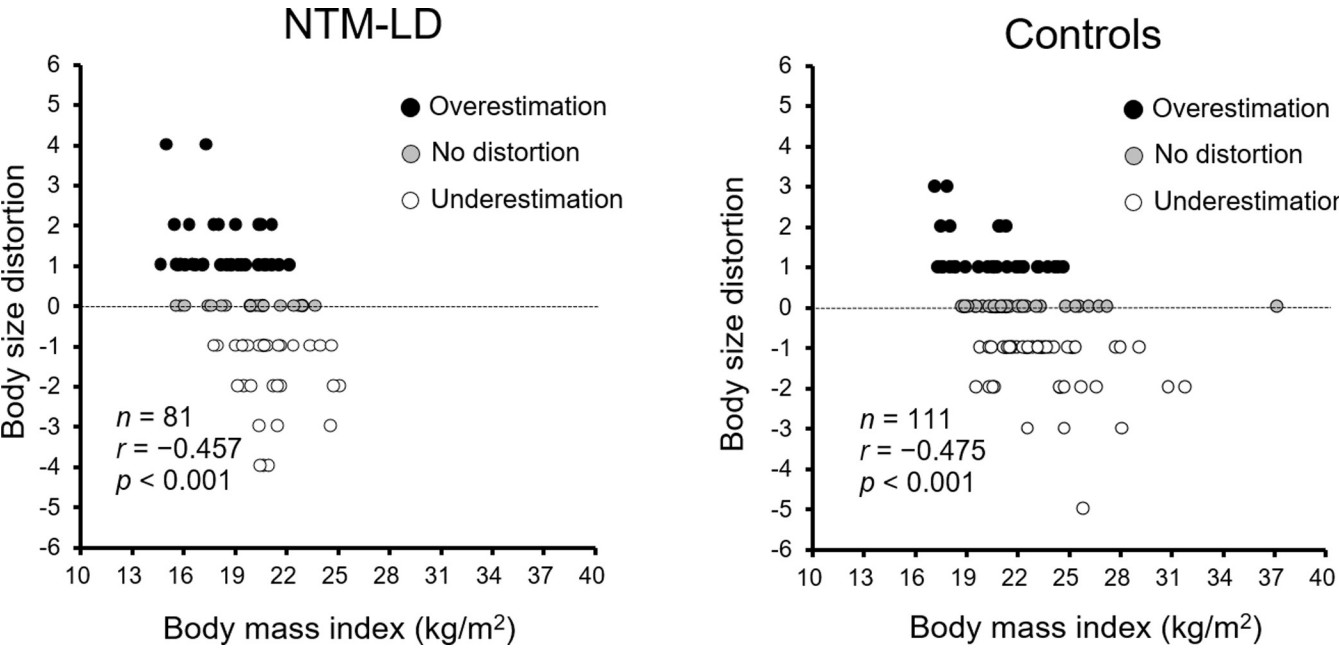

**Fig 1. Correlations between body size distortion and BMI.** *Spearman*'s correlation. Body size distortion was calculated as the difference between the perceived silhouette and the actual silhouette.

## Demographic and clinical data in the OE and non-OE groups

Table 2 shows demographic and clinical data for the OE and Non-OE groups. There were no significant differences in either the demographic or clinical data between the groups.

## Anthropometric and nutritional characteristics in the OE and non-OE patient groups

Table 3 shows the anthropometric measurements, dietary intake, lifestyle indicators, and blood biochemistry data of the patient groups. The OE group had significantly lower body weight, BMI, percentage of body fat ($p < 0.001$ each), and energy ($p = 0.037$) and protein ($p = 0.013$) intakes per 1 kg ideal weight (BMI = 22 kg/m$^2$) [24, 25] than the Non-OE group. In addition, the OE group had a significantly higher longitudinal weight loss compared to the Non-OE group such as a percentage of weight loss from 20 years of age ($p = 0.003$) and from maximum body weight ($p = 0.016$). There were no significant differences in appetite scores, amount of physical activity, and blood biochemistry between the groups. Sensitivity analysis excluding NTM-LD patients with BMI of less than 16.2 kg/m$^2$ yielded similar results for anthropometric data, but for nutritional intake, the significant differences in energy and protein intakes disappeared (S2 Table).

To identify the impact of body size distortion on BMI with adjustment for potential confounders (covariates), we performed multiple regression analyses using BMI as the dependent variable (Table 4). The analysis showed that the duration of NTM-LD ($\beta = -0.171$, 95% CI, $-0.014$ to $-0.001$, $p = 0.027$), percentage of weight loss from age 20 ($\beta = -0.568$, 95% CI, $-0.123$ to $-0.070$, $p < 0.001$), and body size distortion ($\beta = -0.365$, 95% CI, $-0.852$ to $-0.316$, $p < 0.001$) were significantly associated with BMI. These findings indicate that overestimation of body size was associated with lower BMI independent of confounding factors, such as longer disease history, longitudinal weight loss, and nutritional factors. Sensitivity analysis excluding NTM-LD patients with BMI of less than 16.2 kg/m$^2$ showed similar results (S3 Table).

**Table 2. Demographic and clinical data in the OE and the non-OE groups.**

|  | OE ($n = 31$) | Non-OE ($n = 50$) | $p$ value |
|---|---|---|---|
| Age (years) | 70.0 ± 7.3 | 71.3 ± 9.4 | 0.522 |
| Smoking history (packs·year) | 0.03 ± 0.18 | 0.36 ± 1.48 | 0.128 |
| Duration of NTM-LD (month) | 80.3 ± 61.4 | 67.6 ± 52.4 | 0.328 |
| Comorbidity |  |  |  |
| Malignancy (includes lung cancer) | 9 (29.0) | 8 (16.0) | 0.174 |
| Respiratory diseases | 1 (3.2) | 4 (8.0) | 0.644 |
| Diabetes mellitus | 1 (3.2) | 3 (6.0) | 1.000 |
| Cardiac diseases | 3 (9.7) | 5 (10.0) | 1.000 |
| Kidney diseases | 0 (0) | 2 (4.0) | 0.522 |
| Liver diseases | 1 (3.2) | 3 (6.0) | 1.000 |
| NTM causative species |  |  |  |
| *M. avium* | 21 (67.7) | 30 (60.0) | 0.636 |
| *M. intracellulare* | 10 (32.3) | 18 (36.0) | 0.813 |
| *M. kansasii* | 1 (3.2) | 0 (0) | 0.383 |
| *M. abscessus* | 1 (3.2) | 6 (12.0) | 0.242 |

Mean ± standard deviation or n (%), Unpaired t-test (OE vs. Non-OE)

NTM-LD, nontuberculous mycobacterial lung disease

**Table 3. Comparison of anthropometric and nutritional characteristics between the patient groups.**

| | OE ($n$ = 31) | Non-OE ($n$ = 50) | $p$ value |
|---|---|---|---|
| Anthropometric data | | | |
| Height (cm) | 155.8 ± 6.6 | 152.4 ± 6.8 | 0.033 |
| Body weight (kg) | 44.3 ± 5.9 | 48.3 ± 7.6 | 0.014 |
| Body mass index (kg/m$^2$) | 18.2 ± 2.1 | 20.7 ± 2.3 | <0.001 |
| Fat mass (kg) | 10.2 ± 4.1 | 14.4 ± 4.5 | <0.001 |
| Percentage of body fat (%) | 22.6 ± 6.7 | 29.3 ± 6.1 | <0.001 |
| Skeletal muscle mass (kg) | 17.8 ± 2.2 | 17.7 ± 2.7 | 0.902 |
| Weight loss from age 20 (kg) | 5.0 ± 6.9 | 0.0 ± 6.8 | 0.002 |
| Percentage of weight loss from age 20 (%) | 9.5 ± 13.2 | −0.47 ± 14.5 | 0.003 |
| Weight loss from maximum body weight (kg) | 9.5 ± 6.2 | 6.6 ± 5.8 | 0.039 |
| Percentage of weight loss from maximum body weight (%) | 17.3 ± 10.5 | 11.8 ± 9.4 | 0.016 |
| Nutritional intake | | | |
| Energy (kcal/day) | 1,616 ± 310 | 1,709 ± 356 | 0.235 |
| Energy (kcal/IBW1kg/day) [a] | 30.4 ± 6.1 | 33.4 ± 6.4 | 0.037 |
| Protein (g/day) | 59.4 ± 17.4 | 67.3 ± 17.9 | 0.055 |
| Protein (g/IBW1kg/day) [a] | 1.1 ± 0.3 | 1.3 ± 0.3 | 0.013 |
| Fat (g/day) | 47.5 ± 16.0 | 53.5 ± 18.0 | 0.128 |
| Fat (g/IBW1kg/day) [a] | 0.9 ± 0.3 | 1.0 ± 0.3 | 0.053 |
| Carbohydrate (g/day) | 227.6 ± 48.4 | 231.3 ± 60.2 | 0.770 |
| Carbohydrate (g/IBW1kg/day) [a] | 4.3 ± 0.9 | 4.5 ± 1.1 | 0.266 |
| Appetite score (SNAQ-J) | 14.9 ± 1.7 | 14.4 ± 1.6 | 0.208 |
| Amount of physical activity (MET·hour/week) | 29.6 ± 37.7 | 30.5 ± 42.0 | 0.923 |
| Duration of sitting time (hour/day) | 4.4 ± 2.8 | 5.1 ± 2.7 | 0.265 |
| Blood biochemistry | | | |
| Albumin (g/dL) | 4.3 ± 0.3 | 4.4 ± 2.8 | 0.360 |
| Transthyretin (mg/dL) [b] | 19.5 ± 4.9 | 20.3 ± 4.3 | 0.477 |
| Transferrin (mg/dL) | 222 ± 44 | 225 ± 33 | 0.779 |
| Retinol-binding protein (mg/dL) [b] | 2.4 ± 0.6 | 2.6 ± 0.5 | 0.237 |
| Total cholesterol (mg/dL) | 226 ± 33 | 225 ± 38 | 0.943 |
| Hemoglobin (g/dL) | 13.1 ± 0.9 | 13.1 ± 1.3 | 0.937 |
| Lymphocyte count (×10$^2$/μL) | 12.5 ± 4.4 | 14.0 ± 3.7 | 0.084 |

Mean ± standard deviation, Unpaired t-test (OE vs. Non-OE)

[a] Normalized using ideal body weight (BMI = 22 kg/m$^2$).

[b] Data (transthyretin and retinol-binding protein) of one patient in the Non-OE group could not be obtained.

NTM-LD, nontuberculous mycobacterial lung disease; IBW, ideal body weight; SNAQ-J, Japanese version of the Simplified Nutritional Appetite Questionnaire

## Discussion

The main findings of this study were: 1) a strong association between OE and low BMI is shown in the patients as well as in the controls; 2) OE patients had lower BMI, body fat percentage, and energy and protein intakes, and higher weight loss than those who did not; and 3) multiple regression analysis revealed that overestimation of body size was associated with lower BMI independent of confounders, such as longer disease history, longitudinal weight loss, and nutritional factors.

Although previous studies have reported that female patients with NTM-LD are thinner than the general female population [6, 26], this study is the first, to our knowledge, to report the association of BSP with low BMI. In this study, as in the general female population,

**Table 4. Association of BSP with BMI in NTM-LD patients.**

| Variables [a] | B | 95% CI | VIF | p value |
|---|---|---|---|---|
| Body size distortion [b] | −0.365 | −0.852 – −0.316 | 1.262 | <0.001 |
| Age (years) | 0.041 | −0.036 – 0.060 | 1.180 | 0.615 |
| Duration of NTM-LD (month) | −0.171 | −0.014 – −0.001 | 1.027 | 0.027 |
| Percentage of weight loss from age 20 (%) | −0.568 | −0.123 – −0.070 | 1.099 | <0.001 |
| Energy intake (kcal/IBW1kg/day) | 0.105 | −0.024 – 0.106 | 1.258 | 0.215 |
| Appetite score (SNAQ-J) | 0.062 | −0.151 – 0.337 | 1.195 | 0.450 |
| Amount of physical activity (MET·hour /week) | 0.088 | −0.004 – 0.015 | 1.134 | 0.274 |

$n$ = 81. BSP, body size perception; BMI, body mass index; NTM-LD, nontuberculous mycobacterial lung disease; CI, confidence interval; VIF, variance inflation factor; IBW, ideal body weight; SNAQ-J, Japanese version of the Simplified Nutritional Appetite Questionnaire

BMI as the dependent variable

[a] Independent variables

[b] Difference between perceived silhouette and the actual silhouette.

approximately 40% of patients overestimated their body size, and a strong association was found between body size overestimation and low BMI. Previous studies in the general female population have reported an incidence of body size overestimation of 87% in non-obese young Portuguese women [27], 33–60% in non-obese young Taiwanese women [28], and approximately 40–60% in young Asian women [29]. The lower incidence of overestimation in the general female population in the present study than in the early studies may be due to the older age of the participants (> 50 years vs. > 17 years [27–29]), as younger age is more likely to lead to overestimation of body size in women [18]. On the other hand, as many as 40% of overestimation was observed in the female patients in this study, even though their average age was over 70 years. Such overestimation may be caused by factors that are different from those in younger women. Further studies will be necessary for clarification of this issue.

Regarding body size distortion, the negative correlation between body size distortion and actual body size measured by BMI in good agreement with the results of a previous study in Japanese women aged 20–40 years [17]. The present study also showed that body size distortion was associated with the nutritional status in NTM-LD patients and, interestingly, that the OE group had lower energy and protein intakes, and higher weight loss in the long term. These results suggest that the OE group did not consume sufficient energy and nutrients to maintain body weight and, considering their long-term weight loss, they were probably in a negative energy balance. However, the results of the sensitivity analysis excluding patients with a BMI of less than 16.2 kg/m$^2$ showed that the significant differences in energy and protein intakes between the OE and Non-OE groups disappeared, leaving a significant difference only in long-term weight loss. Given the body size distortion and BMI of the healthy controls shown in Fig 1, the frequency of patients with distortion may change significantly if a scale is developed using an expanded control group that includes thinner women. Further studies with the newly developed body image scale are needed to support the present results. Moreover, in female patients with NTM-LD, especially those with the OE, attention should be paid not only to food intake but also to long-term weight loss.

In the multiple regression analysis, body size overestimation was identified as an independent variable significantly associated with the lower BMI in the patients independent of the following confounders: age, duration of the disease, longitudinal weight loss, normalized energy intake, appetite, and amount of physical activity. Previous studies reported that more than half of adult Japanese women perceive their body size as overweight, set their ideal weight at a low

level, and have a strong desire to lose weight [30, 31]. Moreover, women's desire for thinness is reinforced by individual factors, such as the pursuit of beauty and comparison with others, and environmental factors, including mass media and social media [32−35]. Although similar factors may affect female patients with NTM-LD and underlie the association between body size misperception and a lower BMI, it is unknown whether similar trends are observed in NTM-LD patients living in social environments other than Japan. Therefore, further studies are needed. However, the results of this study suggest that the assessment of BSP and its distortion may help in the nutritional management of female patients with NTM-LD to prevent weight loss, and thereby exert a beneficial effect on treatment outcomes. Furthermore, efforts should be made to improve the therapeutic approach for NTM-LD patients and to change the social environment that encourages female thinness in order to prevent the increase of new patients.

In this study, we used the J-BIS [19], which we developed and confirmed its reliability and validity in the Japanese female population, to assess the subjects' BSP. Although there are several studies that have assessed body size perception using the BIS [27, 28, 36], there are some problems as follows: First, the J-BIS was selected based on the subjective evaluation of the participants. As seen in the present results, even healthy controls, especially women with low BMI, may have biased their perception of body size. Moreover, the J-BIS was developed for Japanese women and published in Japanese journal, making it difficult for foreign readers to confirm its reliability and validity or to use the J-BIS. Furthermore, the NTM-LD patients were thinner than the healthy controls, and 10 NTM-LD patients had BMI ($> 16.2$ kg/m$^2$) lower than the BMI range of the slimmest silhouette 1. Although the results excluding these 10 patients were in close agreement with the present results, it is necessary to develop a new BIS that can adequately cover slimmer subjects in the future.

The study has several limitations. First, the height and weight of the control women from the general population were self-reported. However, a previous study in a Japanese population reported sufficient accuracy of self-reported anthropometric measurements [37]. Second, the controls were younger than the NTM-LD patients, so the two groups were not age-matched, but there was no significant correlation between age and body size distortion in the controls. Moreover, the results of additional sensitivity analyses were also consistent with the results of the present study. Therefore, we believe that the controls were suitable for comparing body image and measurement data with NTM-LD patients. Finally, data were collected only from Japanese women; thus, caution should be taken in generalizing the results. However, given the higher incidence of NTM-LD in thin women, our findings may be useful in countries and regions where thinness is prevalent in women.

Despite these limitations, the present study is the first to our knowledge to examine factors associated with thinness in female patients with NTM-LD in terms of BSP. Rather than simply advising these patients to increase their food intake, dietary therapy that also focuses on their BSP may be effective in the treatment and prevention of exacerbation of NTM-LD and merits further investigation in future intervention studies.

## Conclusion

The present findings suggest that body size distortion is an etiological factor for lower BMI in female patients with NTM-LD, supporting our hypothesis that patients' thinness is associated with body size distortion. It may be important to understand patients' body image when providing dietary advice.

## Supporting information

**S1 Table. Anthropometric profiles BSP in NTM-LD and controls with excluding patients BMI less than 16.2 kg/m$^2$.**
(DOCX)

**S2 Table. Comparison of anthropometric and nutritional characteristics between the patient groups with excluding BMI less than 16.2 kg/m$^2$.**
(DOCX)

**S3 Table. Association of BSP with BMI in NTM-LD patients with excluding BMI less than 16.2 kg/m$^2$.**
(DOCX)

## Acknowledgments

The authors thank the patients and medical staff of the Tazuke Kofukai Medical Research Institute, Kitano Hospital. The authors are also grateful to Ms. Natsuki Ueda for collecting dietary data as a registered dietitian.

## Author Contributions

**Conceptualization:** Yumi Takayama, Naoki Sakane, Motonari Fukui, Narumi Nagai.

**Data curation:** Yumi Takayama, Yukina Yumen, Takamasa Kitajima, Noritsugu Honda, Naoki Sakane, Motonari Fukui, Narumi Nagai.

**Formal analysis:** Yumi Takayama, Naoki Sakane, Narumi Nagai.

**Funding acquisition:** Yumi Takayama, Narumi Nagai.

**Investigation:** Yumi Takayama, Yukina Yumen, Takamasa Kitajima, Noritsugu Honda, Motonari Fukui.

**Methodology:** Yumi Takayama, Motonari Fukui, Narumi Nagai.

**Supervision:** Motonari Fukui, Narumi Nagai.

**Validation:** Naoki Sakane, Narumi Nagai.

**Writing – original draft:** Yumi Takayama, Narumi Nagai.

**Writing – review & editing:** Yukina Yumen, Takamasa Kitajima, Noritsugu Honda, Naoki Sakane, Motonari Fukui, Narumi Nagai.

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
