## [Decision Letter · Decision Letter 0]

24 Jan 2023

PONE-D-22-32070Association of body size perception with body mass index, long-term energy balance, and nutritional status in female patients with pulmonary nontuberculous mycobacterial lung diseasePLOS ONE

Dear Dr. Takayama,

Thank you for submitting your manuscript to PLOS ONE. After careful consideration, I feel that it has merit but does not fully meet PLOS ONE’s publication criteria as it currently stands. Especially, one reviewer showed his serious concerns in statistical analysis and methodology. I have added a number of further suggestions in the statistical analysis and methodology myself.

I invite you to respond to the reviewers' comments and my own, and submit a revision if you believe you can adequately address the concerns. Please note that this request for revision does not imply that the manuscript will ultimately be accepted, but that the manuscript must be meticulously revised before further consideration is provided.

We look forward to receiving your revised manuscript.

Kind regards,

Yuichiro Nishida

Academic Editor

PLOS ONE

Journal Requirements:

"This study was supported by grants from the Kitano Research Incentives and Respiratory Disease Center, Tazuke Kofukai Medical Research Institute, Kitano Hospital. This work was supported in part by MHLW Comprehensive Research Project against Cardiovascular Diseases, Diabetes and Other Lifestyle Related Diseases Program Grant Number JPMH 22FA1023."

Additional Editor Comments:

MAJOR:

Abstract (Methods): Important descriptions on the study design (cross-sectional), how to recruit control women (web survey), and the main statistical analysis (multiple regression analysis) are all missing in the Abstract. Please be explicit about what is main independent variable and a dependent variable in the multiple regression analysis.

Introduction: Overall (not only the Introduction section but also the other sections), the manuscript is written in poor English. The manuscript should be checked by a native speaker.

Line 112: Height and weight in control women are self-reported (not objectively measured). Thus, BMI data in the control group are based on self-reported height and weight. Please be explicit about that point throughout the manuscript.

Lines 156-159: There is almost no information on the questionnaires used in the present study. Citing your previous study is insufficient, brief explanations on the questionnaires are needed. For instance, which questionnaire was used to assess dietary intake? How did you calculate energy, protein, fat, carbohydrate intakes. Is the diet questionnaire used in the present study validated? Did you use validated questionnaires for assessing physical activity and appetite score? What about smoking history?

Line 174: In the footnotes of Tables 1-3, it is described that UNPAIRED t-test was used. However, you mentioned paired Student’s t-test or Welch's t-test in this line.

Line 176: Please be explicit about which variable (BMI or body size distortion) is an independent variable or a dependent variable. What is the purpose of this analysis? Which hypothesis is tested by performing the univariate analysis.? Please explain courteously the reason why you performed this analysis. Or consider leaving it out. I can understand if you use body size distortion as an independent variable (since the body size distortion is a potential cause) and BMI as a dependent variable (since BMI is an outcome), as you did in the followed multiple regression analysis (as shown in Table 4) in order to examine your initial hypothesis described in the Introduction section. However, I could not clearly understand your intent for performing the univariate analysis, in which BMI is treated as an independent variable, and body size distortion is treated as a dependent variable.

Lines 180-184: As Reviewer 1 pointed out, it is critically important that this sentence should be rephrased to “impact of age etc (explanatory or independent variables) on BMI (outcome or dependent variables)”.

In the Introduction section (lines 98-100), you mentioned that you examined the impact of body size overestimation on their body size and nutritional status. To examine this, you should treat the body size overestimation (potential cause) as an independent variable and the body size (BMI) and nutritional status (potential outcomes) as dependent variables in the association analysis. However, you treat energy intake (nutritional status) as an independent variable in the multiple regression analysis (results show in Table 4). Why?

You treated the body size distortion as a main independent variable in the multiple regression analysis. Please clearly explain whether you treat other six independent variables (age, duration of NTM-LD, percentage of weight loss from age 20, etc) as potential confounding factors or mediating factors. Usually, mediating factors should not be included as covariates in multivariate regression models, since the inclusion of mediating factors into multivariate models disappears the significant association between a main independent variable and an outcome.

Lines 267: Throughout the results section, please be explicit about which variable is an independent variable or a dependent variable.

Line 277: Discussion must be appropriately revised according to the changes you made in the Methods and Results sections according to reviewers’ and my comments.

MINOR:

Line 87: full name first, abbreviation second.

Lines 256-258: β values are missing.

Reviewers' comments:

Reviewer's Responses to Questions

**Comments to the Author**

1. Is the manuscript technically sound, and do the data support the conclusions?

Reviewer #1: Yes

Reviewer #2: Partly

2. Has the statistical analysis been performed appropriately and rigorously? 

Reviewer #1: Yes

Reviewer #2: No

3. Have the authors made all data underlying the findings in their manuscript fully available?

Reviewer #1: Yes

Reviewer #2: Yes

4. Is the manuscript presented in an intelligible fashion and written in standard English?

Reviewer #1: Yes

Reviewer #2: Yes

5. Review Comments to the Author

Reviewer #1: This study is useful in providing greater insight into some reasons underlying the lower BMI among Japanese women with NTM lung disease, namely a distorted body perception, as well as the factors predictive of low BMI in this same population. The study is thorough, and adds important information to the field. In particular, the role of low BMI in disease susceptibility is clear ,and the difficult and often complicated factors underlying the low BMI and the need for improved intervention are highlighted in this paper.

Minor comments for improving this manuscript are listed below.

1) Suggest finding a term other than “elderly”, as this term is subjective; suggest instead explicitly using an age value

2) Find a better term than “ordinary”- suggest “controls”. Please be consistent in referring to “controls” rather than “ordinary”

3) Results

o Controls were on average 10 years older than cases… was it not possible to find controls within the same general age range? Ie frequency match?

4) Be consistent in the labelling of the OE vs Remaining….. maybe call these “non OE” or controls- see comment above

5) Remove the “all” column… not sure what that adds, as the primary comparison is between cases and controls.

6) Lines 181-183: “ impact of BMI (response variable) on age, etc”: this should be rephrased to “impact of age etc (these are the explanatory variables) on BMI (response variable)

7) Confusing to say “correlation between body size perception and BMI in NTM LD and controls”… .. since you have to have a control group to make these comparisons… suggest just saying “correlation between body size distortion and BMI

Reviewer #2: Overview:

The authors conducted a study about body size perception in patients with nontuberculous mycobacterial lung disease (NTM-LD). I agree that it is important to address the problem of nutritional status of patients with NTM-LD because low BMI is associated with poor prognosis. While I appreciate the effort of this study which included more than 80 female patients, there are some questions about the methodology of the study.

Major comments:

1. Evaluation of body size perception

The main concern I have about this study is evaluation of body size perception (BSP). To assess patients' BSP, the authors used the Japanese version of body image scale they had developed. This scale was developed using the subjective responses and BMI from a healthy population (in which BMI was not evenly distributed), rather than using an image created from objective data of physical measurements. On this scale, the (mean) BMIs associated with low-numbered silhouettes seem to tend to shift to higher values.

When using this scale to analyze populations with very low BMI, doesn’t it affect the assessment of overestimation of one’s body size (OE)?

In fig1, there seems to be a certain number of NTM-LD patients with a BMI <16. Patients with a BMI <16 cannot be “underestimation (BSP <0)”, because the range starts at silhouette 1 = BMI 16.2 on this scale. This affected the distribution of dots in fig1. For this reason, BSP may have been more strongly associated with BMI in multiple regression analysis (table 4).

Isn't it difficult to accurately assess BSP without using an “objective” scale that extends the range of lower BMI?

2. BMI cutoffs for overestimation

Did the authors use the automatically calculated cutoff (Youden index?) in any of the "R" packages?

The BMI cutoff value for NTM-LD seems reasonable from the ROC (fig2). For the control, the BMI cutoff value from the ROC (fig2) seems to vary depending on the method. If a similar sensitivity (or specificity) to NTM-LD is chosen, the BMI cutoff value for the control is likely to be lower. In this study, it seems that there is no statistical meaning in examining the absolute value of the difference in the automatically calculated cutoff.

Minor comments:

line 31, line 70: Remove "pulmonary".

line 133: Please provide details on how to convert BMI to silhouette number (actual silhouette).

6. PLOS authors have the option to publish the peer review history of their article (what does this mean?). If published, this will include your full peer review and any attached files.

Reviewer #1: No

Reviewer #2: No

---

## [Author Response · Author response to Decision Letter 0]

23 Mar 2023

The editor: I am grateful to the editor and reviewers for helpful suggestions. As indicated in the uploaded the document as a “Response to reviewers” files, I have taken all the suggestions into account in the revised manuscript.

Reviewer 1: I have incorporated all of your suggestions into my revised manuscript. They were very helpful. Thank you. 

Please see an uploaded the document as a “Response to reviewers” file.

Reviewer 2: I have incorporated all of your suggestions into my revised manuscript. They were very helpful. Thank you. 

Please see an uploaded the document as a “Response to reviewers” file.

---

## [Decision Letter · Decision Letter 1]

17 May 2023

PONE-D-22-32070R1Association of low body mass index with body size distortion in female patients with nontuberculous mycobacterial lung diseasePLOS ONE

Dear Dr. Takayama,

Thank you for submitting your manuscript to PLOS ONE. After careful consideration, we feel that it has merit but does not fully meet PLOS ONE’s publication criteria as it currently stands. Therefore, we invite you to submit a revised version of the manuscript that addresses the points raised during the review process.

We look forward to receiving your revised manuscript.

Kind regards,

Yuichiro Nishida

Academic Editor

PLOS ONE

Additional Editor Comments:

Major comments:

Title: You misunderstood my comment, and you made an unwanted correction. The title should be expressed as “Association of body size distortion with low body mass index in female patients with nontuberculous mycobacterial lung disease”. I have never seen an expression such as “Association of [dependent variable] with [independent variable]” in scientific literature. To prevent readers’ confusion, you should express the “Association of [independent variable] with [dependent variable]” throughout the manuscript.

Abstract; Page 2, lines 33-34: When you mention your hypothesis, you should use words, such as “would be” or “might be”. Your explanation “their thinness was associated with a misperception of body shape” is like a result (like a fact, not a hypothesis).

Abstract: Your descriptions regarding the multiple regression analysis are inconsistent in the Background and Method. In the Background, you mentioned that “To test this hypothesis, we examined whether the patients’ body mass index (BMI) was related to their body size perception (BSP), independent of age, disease duration, and nutritional factors.” However, in the Methods, you mentioned that “Multiple regression analysis was performed to identify factors independently associated with patients' BMI" which is totally different from the purpose of testing the above-mentioned hypothesis. Based on your descriptions in the Abstract (Background) and last paragraph of the Introduction section (lines 94-98), your purpose of the present study can be NOT to identify factors to be independently associated with BMI, but the current purpose is to clarify whether BSP is associated with BMI independently from potential confounding factors. Your inconsistent description in the Background and Methos can confuse readers a lot. For testing your hypothesis, your description on the multiple regression analysis in the Method section should be expressed as “Multiple regression analysis was performed to clarify the association between BSP (independent variable) and BMI (dependent variable) with adjustment for potential confounders, such as age, disease duration, and nutritional factors”. Accordingly, the Results section should also be appropriately revised. For instance, it may be expressed as “multiple regression analysis revealed that overestimation of body size was associated with lower BMI independent of confounding factors, such as longer disease history nutritional factors, and longitudinal weight loss”.

Page 5, lines 96-98: If you mentioned “BSP was related to BMI (independent variable comes first, followed by the dependent variable)”, it may not cause readers’ confusion. You should add explanations with parentheses like “patients' BMI (dependent variable) was related to their BSP (independent variable)” to prevent readers’ confusion.

Page 9, lines 182-185: As mentioned above, based on your descriptions in the Abstract (Background) and last paragraph of the Introduction section (lines 94-98), the purpose of the present study is to clarify whether BSP is associated with BMI independently from potential confounding factors. Thus, this explanation of the multiple regression analysis is wrong. According to the purpose of the current study, you should mention that “Multiple regression analysis was performed to evaluate the impact of body size distortion (independent variables) on BMI (dependent variable) with adjustment for potential confounding factors (covariates), such as age, duration of NTM-LD, percentage of weight loss from 20 years of age, normalized energy intake, appetite score, and physical activity.

Page 14, Table 4: It is OK to show the results (β values, their CI, and P values) of confounding factors such as age, but the results (β values, their CI, and P values) for the most important variable, body size distortion, should be shown in the first top line (not the last or bottom line) of Table 4. The title may be revised to “Association of BSP with BMI in NTM-LD patients”.

Page 9, lines 185-187: You misunderstood my previous comment on the confounding or mediating factors, and you made an unwanted correction. Whether a certain variable is treated as a confounding factor or mediating factor should be determined before performing statistical analysis, based on the authors’ interest (and authors’ hypothesis), but not based on the results of the mediation analysis. For instance, if you are interested in testing a hypothesis that the effect of BSP (independent variable) on BMI (dependent variable) would be mediated by total energy intake, it is appropriate to treat the variable (total energy intake) as a potential mediating factor. Thus, in this case, it is also appropriate to perform a mediation analysis. Since it was unclear to me whether you are interested in “the mediation” in the previous version of the manuscript, I thus asked whether you are treating variables as confounding factors or mediating factors. Based on the present version of the manuscript, it appears that you are not interested in any mediation, and do not have any hypothesis regarding mediation. Thus, you should not perform any mediation analysis in the present study. You should delete all the descriptions related to mediation analysis from the manuscript.

The results and discussion sections should also be carefully revised according to the new revisions regarding the statistical analyses.

Overall, I agree with Reviewer 2’s comments.

I understand that you can't do anything about having used the Japanese version of the body image scale that you own developed. However, you may be able to emphasize the most important limitations (pointed out by Reviewer 2) more clearly, meticulously, and courteously, by making one new paragraph (separately from the limitation paragraph) in the Discussion section. I understand that you performed the sensitivity analyses with the exclusion of patients whose BMI < 16.2, as a last resort (Kunikunosaku in Japanese). However, I could not find any descriptions regarding the sensitivity analyses (in the Methods or Results sections, or Tables) in the revised manuscript. Your revision according to the precious comments from Reviewer 2 has not been incorporated into the revised manuscript. I suggest you show the results of sensitivity analyses as supplemental files. A thorough discussion on the results of sensitivity analyses is needed. Unfortunately, your response to the Reviewer 2’s comment on the BMI cutoff values is also insufficient.

Reviewers' comments:

Reviewer's Responses to Questions

**Comments to the Author**

1. If the authors have adequately addressed your comments raised in a previous round of review and you feel that this manuscript is now acceptable for publication, you may indicate that here to bypass the “Comments to the Author” section, enter your conflict of interest statement in the “Confidential to Editor” section, and submit your "Accept" recommendation.

Reviewer #1: All comments have been addressed

Reviewer #2: (No Response)

2. Is the manuscript technically sound, and do the data support the conclusions?

Reviewer #1: Yes

Reviewer #2: Partly

3. Has the statistical analysis been performed appropriately and rigorously? 

Reviewer #1: Yes

Reviewer #2: Yes

4. Have the authors made all data underlying the findings in their manuscript fully available?

Reviewer #1: Yes

Reviewer #2: No

5. Is the manuscript presented in an intelligible fashion and written in standard English?

Reviewer #1: Yes

Reviewer #2: Yes

6. Review Comments to the Author

Reviewer #1: (No Response)

Reviewer #2: Major comments

1) Evaluation of body size perception

I still think that it is important to define a standard body image to evaluate body size perception. Unfortunately, readers outside Japan will not be able to confirm the validity of the scale used in this study.

As the authors describe, even healthy controls (especially with low BMI) can have body size distortion. Therefore, in order to assess BSD properly, it seems necessary to use a scale developed by the objective indicators rather than a scale developed by the subjective judgment of controls. The use of authors’ scale seems to be a major limitation of this study when assessing “overestimation of body size”. Shouldn't the authors clearly state the issue?

In this paper, the authors seem to overemphasize the frequency of patients with OE (∼40%). Considering body size distortion and BMI in the controls shown in fig 1, the frequency of patients with distortion could vary significantly if a scale is developed using the controls in this study (or the expanded control set).

In addition, though there is no significant difference between the frequency of OE in the patients with NTM-LD and the controls, the authors seem to emphasize the higher frequency of OE in the patients (line 195, line 277...).

As authors hypothesized (line 90), the female patients with NTM-LD have body size distortion similar to that of the general adult female population. In this study, a strong association between low BMI and OE is shown in the patients as well as in the controls.

Sensitivity analysis

Authors performed a sensitivity analysis on Figure 1 and Table 4, excluding 10 patients with BMI less than 16.2 kg/m2. Sensitivity analysis showed that the patients (r=-0.392) had a weaker correlation between BMI and BSP than the controls (r=-0.497).

How about the frequency of “overestimation of body size” in table 1 and “Nutritional intake” in table 3 when excluding the patients with BMI less than 16.2 kg/m2?

2) BMI cutoffs for overestimation

●Revised Results (line 211- 212):

“BMI cutoff values in the NTM-LD (19.2 kg/m2) patients that overestimate their body size were lower than the controls (22.3 kg/m2).”

Again, how did the authors calculate BMI cutoff values?

Without knowing the sensitivity and specificity for each cutoff value, it is difficult to compare values.

7. PLOS authors have the option to publish the peer review history of their article (what does this mean?). If published, this will include your full peer review and any attached files.

Reviewer #1: No

Reviewer #2: No

---

## [Author Response · Author response to Decision Letter 1]

25 Jun 2023

"Response to the reviewers" has been uploaded as a file.

---

## [Decision Letter · Decision Letter 2]

11 Jul 2023

PONE-D-22-32070R2Association of body size distortion with low body mass index in female patients with nontuberculous mycobacterial lung diseasePLOS ONE

Dear Dr. Takayama,  Thank you for submitting your manuscript to PLOS ONE. After careful consideration, we feel that it has merit but does not fully meet PLOS ONE’s publication criteria as it currently stands. Therefore, we invite you to submit a revised version of the manuscript that addresses the points raised during the review process. Please submit your revised manuscript by Aug 25 2023 11:59PM. If you will need more time than this to complete your revisions, please reply to this message or contact the journal office at plosone@plos.org. Please include the following items when submitting your revised manuscript:A rebuttal letter that responds to each point raised by the academic editor and reviewer(s). You should upload this letter as a separate file labeled 'Response to Reviewers'.A marked-up copy of your manuscript that highlights changes made to the original version. You should upload this as a separate file labeled 'Revised Manuscript with Track Changes'.An unmarked version of your revised paper without tracked changes. You should upload this as a separate file labeled 'Manuscript'.If applicable, we recommend that you deposit your laboratory protocols in protocols.io to enhance the reproducibility of your results. Protocols.io assigns your protocol its own identifier (DOI) so that it can be cited independently in the future. For instructions see: https://journals.plos.org/plosone/s/submission-guidelines#loc-laboratory-protocols. Additionally, PLOS ONE offers an option for publishing peer-reviewed Lab Protocol articles, which describe protocols hosted on protocols.io. Read more information on sharing protocols at https://plos.org/protocols?utm_medium=editorial-email&utm_source=authorletters&utm_campaign=protocols.

We look forward to receiving your revised manuscript.

Kind regards,

Yuichiro Nishida

Academic Editor

PLOS ONE

Journal Requirements:

**Additional Editor Comments:**

To be consistent with Table 1, a superscript "a" of p value should be moved to the variables (age, height, body weight, and body mass index) in S1 Table.

Reviewers' comments:

Reviewer's Responses to Questions

**Comments to the Author**

1. If the authors have adequately addressed your comments raised in a previous round of review and you feel that this manuscript is now acceptable for publication, you may indicate that here to bypass the “Comments to the Author” section, enter your conflict of interest statement in the “Confidential to Editor” section, and submit your "Accept" recommendation.

Reviewer #2: (No Response)

2. Is the manuscript technically sound, and do the data support the conclusions?

Reviewer #2: Partly

3. Has the statistical analysis been performed appropriately and rigorously? 

Reviewer #2: Yes

4. Have the authors made all data underlying the findings in their manuscript fully available?

Reviewer #2: No

5. Is the manuscript presented in an intelligible fashion and written in standard English?

Reviewer #2: Yes

6. Review Comments to the Author

Reviewer #2: Comments

BMI cutoffs for overestimation

line 184: “Diagnostic accuracy using the area under the ROC curve was defined as follows; < 0.5 (test not useful), 0.5-0.6 (poor), 0.6-0.7 (sufficient), 0.7-0.8 (good), 0.8-0.9 (very good), and 0.9-1.0 (excellent) using the R software package.”

The values of area under the ROC curve (AUC) are not described in the results section of the revised manuscript.

line 224: “BMI cutoff values in the NTM-LD (19.2 kg/m2, Sensitivity = 0.800, Specificity = 0.677) patients that overestimate their body size were lower than the controls (22.3 kg/m2, Sensitivity = 0.500, Specificity = 0.857).”

Looking at fig2 of the first version of the paper, it seems that sensitivity and specificity are mixed up in the revised paper. (Please check for any mistakes. Sensitivity = 0.677, Specificity = 0.800 for NTM-LD? and Sensitivity = 0.857, Specificity = 0.500 for the controls?)

If you mention that the cutoff value is lower in the NTM-LD patients, you should also mention that the sensitivity is lower in the patients. I think that there is little significance in comparing cut-off values with different sensitivities.

7. PLOS authors have the option to publish the peer review history of their article (what does this mean?). If published, this will include your full peer review and any attached files.

Reviewer #2: No

---

## [Author Response · Author response to Decision Letter 2]

25 Jul 2023

"Response to the reviewers" has been uploaded as a file.

---

## [Editor Report · Decision Letter 3]

28 Jul 2023

PONE-D-22-32070R3Association of body size distortion with low body mass index in female patients with nontuberculous mycobacterial lung diseasePLOS ONE

Dear Dr. Takayama,

Thank you for submitting your manuscript to PLOS ONE. After careful consideration, we feel that it has merit but does not fully meet PLOS ONE’s publication criteria as it currently stands. Therefore, we invite you to submit a revised version of the manuscript that addresses the points raised by the editor.

We look forward to receiving your revised manuscript.

Kind regards,

Yuichiro Nishida

Academic Editor

PLOS ONE

Journal Requirements:

Additional Editor Comments (if provided):

Page 9, lines 179-185: I have recognized that the explanation of ROC curve analysis is insufficient. Readers will be confused a lot, because even the purpose of the ROC curve analysis is not clearly described, and the methodology of ROC curve analysis is inappropriately described in the Methods section. Please consider the following descriptions: The receiver operating characteristic (ROC) curves were drawn to assess the prediction performance regarding the BMI and body size overestimation (OE) in the NTM-LD and control groups, and their area under the curves (AUC) were calculated using ROC curves. To compare the optimal cut-off values of BMI to best predict the body size OE between the two groups, the optimal cut-off values were calculated by Youden index, and the sensitivity and specificity under the optimal cut-off points in the ROC curves were obtained.

Although these suggested descriptions may not be correct, you can use them as a reference for your appropriate revision. Again, please explain the purpose and method of the ROC curve analysis more clearly and accurately for readers.

Page 11, line 219: This sentence may also not be correct. Consider the following sentence “The optimal BMI cut-off values in the NTM-LD patients (19.2 kg/m2, AUC = 0.784, Sensitivity = 0.677, Specificity = 0.800) best predicting the overestimation of body size were ……”.
---

## [Author Response · Author response to Decision Letter 3]

3 Aug 2023

"Response to reviewers" has been uploaded as a file.

---

## [Editor Report · Decision Letter 4]

4 Aug 2023

Association of body size distortion with low body mass index in female patients with nontuberculous mycobacterial lung disease

PONE-D-22-32070R4

Dear Dr. Takayama,

We’re pleased to inform you that your manuscript has been judged scientifically suitable for publication and will be formally accepted for publication once it meets all outstanding technical requirements.

Kind regards,

Yuichiro Nishida

Academic Editor

PLOS ONE

Additional Editor Comments (optional):

Congratulations!

---

## [Editor Report · Acceptance letter]

11 Aug 2023

PONE-D-22-32070R4 

Association of body size distortion with low body mass index in female patients with nontuberculous mycobacterial lung disease 

Dear Dr. Takayama:

I'm pleased to inform you that your manuscript has been deemed suitable for publication in PLOS ONE. Congratulations! Your manuscript is now with our production department. 

Kind regards, 

on behalf of

Dr. Yuichiro Nishida 

Academic Editor

PLOS ONE